# Impact of Body Composition Status on 90-Day Mortality in Cancer Patients with Septic Shock: Sex Differences in the Skeletal Muscle Index

**DOI:** 10.3390/jcm8101583

**Published:** 2019-10-02

**Authors:** Youn-Jung Kim, Dong-Woo Seo, Jihoon Kang, Jin Won Huh, Kyung Won Kim, Won Young Kim

**Affiliations:** 1Department of Emergency Medicine, University of Ulsan College of Medicine, Asan Medical Center, Seoul 05505, Korea; yjkim.em@gmail.com (Y.-J.K.); leiseo@gmail.com (D.-W.S.); 2Department of Biomedical Informatics, University of California San Diego, School of Medicine, La Jolla, CA 92093, USA; 3Department of Hematology/Oncology, Department of Internal Medicine, Kangbuk Samsung Medical Center, Sungkyunkwan University School of Medicine, Seoul 03181, Korea; jihoon_kang@hotmail.com; 4Department of Pulmonary and Critical Care Medicine, University of Ulsan College of Medicine, Asan Medical Center, Seoul 05505, Korea; jwhuh@amc.seoul.kr; 5Department of Radiology, University of Ulsan College of Medicine, Asan Medical Center, Seoul 05505, Korea

**Keywords:** septic shock, neoplasms, prognosis, body composition, sarcopenia

## Abstract

Abnormalities in body composition are associated with poor prognosis in cancer patients. We investigated the association between body composition and 90-day mortality in cancer patients who developed septic shock. We included consecutive septic shock patients with active cancer from 2010 to 2017. The muscle area at the level of the third lumbar vertebra was measured by computed tomography upon emergency department admission and adjusted by height squared, yielding the Skeletal Muscle Index (SMI). Hazard ratios (HRs) and 95% confidence intervals (CIs) for 90-day mortality were estimated using a Cox proportional hazards model. Among 478 patients, the prevalence of muscle depletion was 87.7%. Among markers of body composition, the SMI only differed significantly between non-survivors and survivors (mean, 35.48 vs. 33.32 cm^2^/m^2^; P = 0.002) and was independently associated with lower 90-day mortality (adjusted HR, 0.970; P = 0.001). The multivariable-adjusted HRs (95% CI) for 90-day mortality comparing quartiles 2, 3, and 4 of the SMI to the lowest quartile were 0.646 (0.916–1.307), 0.620 (0.424–0.909), and 0.529 (0.355–0.788), respectively. The associations were evident in male patients, but not in female patients. The SMI was independently associated with 90-day mortality in cancer patients with septic shock. The graded association between the SMI and 90-day mortality was observed in male patients.

## 1. Introduction

Recent advances in cancer treatment, including therapeutic interventions and supportive care, have improved the overall survival rates of patients with cancer [1]. The impairment of cellular and humoral immune systems due to cancer progression, therapeutic interventions such as chemo-radiation-therapy, and malnutrition are common in patients with cancer and increase vulnerabilities to infection, which can lead to rapid progression to sepsis and septic shock, a life-threatening or even fatal complication [2,3,4,5].

Recently, body composition in patients with cancer has been suggested as an important feature to predict the tolerance to therapeutic interventions as well as clinical outcomes, including survival and functional status before and during treatment [6,7,8,9,10]. Sarcopenia, defined as muscle depletion and decreased muscle function, is a reversible and preventable condition with adequate therapeutic interventions such as nutrition and physical rehabilitation [11]. Both tumor- and host-derived factors such as chronic inflammatory responses and a hypercatabolic state contribute to ongoing deterioration in patients with cancer and abnormalities of body composition have been suggested as another target requiring therapeutic intervention to improve survival and quality of life [12,13,14].

Several studies demonstrated the association between body composition, including muscle mass, sarcopenia, or visceral fat and mortality in patients with sepsis or in patients with cancer; however, the impact of body composition on the outcome of the patients with cancer who develop septic shock is not yet elucidated [7,10,15,16,17,18]. We hypothesized that body composition is a powerful prognostic factor for patients with cancer who develop septic shock. Thus, this study assessed the body composition status, including muscle area index, skeletal muscle attenuation, visceral/subcutaneous fat area, muscle depletion and obesity in cancer patients who presented to the emergency department (ED) with septic shock and evaluated the association between body composition and 90-day mortality.

## 2. Materials and Methods

### 2.1. Study Design and Population

The institutional review board of our hospital approved the registry (IRB number: 2015-1253), and informed consent was obtained before data collection. This observational, prospectively collected registry-based study was conducted at the ED of a tertiary-care university-affiliated hospital in Seoul, Korea, with an annual census of approximately 110,000 visits. Since 2010, all adult (≥18 years) patients with suspected or confirmed infection and evidence of refractory hypotension or hypoperfusion at ED are enrolled in the septic shock registry of our center [19]. Refractory hypotension was defined as persistent hypotension (systolic blood pressure <90 mmHg, a mean arterial pressure <70 mmHg, or a systolic blood pressure decrease >40 mmHg) after administering ≥20–30 mL/kg intravenous fluid or requiring vasopressors to maintain a systolic blood pressure of ≥90 mmHg or mean arterial pressure of ≥70 mmHg [19]. Hypoperfusion was defined as a serum lactate level ≥4 mmol/L [20]. Patients were excluded in our septic shock registry if they refused admission to the intensive care unit and intensive treatment, signed a “do not attempt resuscitation” order, developed septic shock 6 or more hours after ED arrival, were transferred from other hospitals after stabilization, were directly transferred to other hospitals from the ED, or refused to enroll in the registry.

This study included patients with active solid cancer who were enrolled in the septic shock registry between January 2010 and December 2017 and who underwent abdominopelvic computed tomography (CT) examination at ED presentation. Active solid cancer was defined as a solid tumor which had been diagnosed or treated within the past 6 months, or with non-resectable and distant metastasis [21]. Patients were categorized into 90-day survivor and non-survivor groups. The institutional review board of our hospital approved this study (IRB number: 2019-0147) and waived the requirement for informed consent because the additional data were retrospectively retrieved from electronic medical records.

All patients were treated with sufficient crystalloid administration, with monitoring of volume status, acquisition of blood cultures before antibiotic administration, and administration of broad-spectrum antibiotics immediately after blood cultures and vasopressors in accordance with the then-current guidelines and bundles of the Surviving Sepsis Campaign [19]. Diagnostic work-up, including laboratory examination and computed tomography (CT) scans, were also performed in the ED. Patients were followed up until 90 days after hospital admission or the time of death.

### 2.2. Data Collection

Data regarding age, sex, comorbid disease, focus of infection, sequential organ failure assessment (SOFA) score, and 90-day mortality were retrieved from the registry. SOFA score was calculated using the worst parameters during the initial 24 h after ED admission. The primary endpoint was 90-day mortality. 

Additional data were collected for this study using the electronic medical records, including weight, height, type of solid cancer, and the presence of abdominopelvic CT scan to evaluate the body composition. We retrieved the data of body weight and height entered into the electronic health records at admission. The body composition, including skeletal muscle area (SMA), visceral fat area (VFA), and the subcutaneous fat area (SFA), was assessed at the L3 vertebral level of abdominopelvic CT scan performed at ED presentation. Abdominopelvic CT images extending from L3 in the inferior direction, which was performed at ED presentation, were assessed. An experienced radiologist (K.W.K) analyzed the CT images using AsanJ-Morphometry ^TM^ software. This dedicated software measures abdominal muscle and fat area based on ImageJ (NIH, Bethesda, MD, USA) [22]. The SMA was demarcated using predetermined thresholds (−29 to +190 Hounsfield units [HU]); the VFA and SFA were also demarcated using fat tissue thresholds (−190 to −30 HU) [23]. Skeletal muscle attenuation was assessed as the mean radiodensity in HU of all SMA at L3.

Body mass index (BMI) was calculated as the weight in kilograms divided by the height squared in meters (kg/m^2^). The Skeletal Muscle Index (SMI) was calculated as the SMA in cm^2^ divided by the height squared in meters (cm^2^/m^2^) [7]. Obesity was defined as a BMI of 25 kg/m^2^ or higher, which is the proposed cutoff for a diagnosis of obesity in Asians [24]. Sarcopenia, diagnosed based on the low muscle mass and impaired function, was defined using sex-specific SMIs [25]. Although the SMI cut-off values were proposed differently, we used sex-specific, BMI-dependent SMI cutoffs [7,18]. Sarcopenia was defined as <43 cm^2^/m^2^ for a BMI < 25 kg/m^2^, <53 cm^2^/m^2^ for a BMI of 25 kg/m^2^ or more, and <41 cm^2^/m^2^ regardless of the BMI for female patients in a previous study [18]. In addition, due to the lack of data about muscle strength or physical performance, we defined muscle depletion according to the previous sex-specific, BMI-dependent SMI cutoffs. Low skeletal muscle attenuation was defined as <41 HU for a BMI < 25 kg/m^2^ and <33 HU for a BMI of 25 kg/m^2^ or more [18].

### 2.3. Statistical Analysis

Descriptive statistics were used to summarize the characteristics of the study patients according to 90-day mortality and according to the presence of muscle depletion. Continuous variables are presented as means (standard deviation, SD) or medians (interquartile range, IQR) according to their distribution per the Kolmogorov–Smirnov test. Categorical variables are presented as absolute numbers (percentage). Sex-specific SMIs were grouped into quartiles based on the distribution within the study patients as follows: <31.03, 31.03–36.45, 36.46–42.06, and >42.06 kg/m^2^ for male and <27.22, 27.22–31.35, 31.36–35.47, and >35.47 kg/m^2^ for female patients. Additionally, odds ratios (ORs) and 95% confidence intervals (CIs) for the presence of muscle depletion were examined using univariable and multivariable logistic regression analysis. Variables were tested for goodness of fit using variable methods such as the Hosmer–Lemeshow test and Stukel test.

The primary endpoint was all-cause 90-day mortality. Each participant was followed from ED admission until either death or 90 days after ED admission, whichever occurred first. Hazard ratios (HRs) and 95% CIs for 90-day mortality were estimated using Cox proportional hazards regression analyses. The model was adjusted for age, sex, and other variables that might affect 90-day mortality: hypertension, diabetes mellitus, type of solid cancer, focus of infection, SOFA score at admission, the SMI and skeletal muscle attenuation. The proportional hazards assumption was assessed by examining graphs of the estimated log minus log plots and no violation of the assumption was found. We also evaluated the association between the SMI and 90-day mortality separately in female and male patients. Survival in the sex-specific SMI quartiles was assessed by nonparametric Kaplan–Meier survival analysis and compared by log-rank tests. Two-tailed P-values less than 0.05 were considered statistically significant. All statistical analyses were performed using IBM SPSS Statistics for Windows, version 20.0 (IBM Corp., Armonk, NY, USA).

## 3. Results

During the study period, a total of 2425 patients with septic shock were enrolled in the septic shock registry and 478 (19.7%) patients with active solid cancer were finally included. Among these 478 patients, 208 (43.5%) died within 90 days (Figure 1).

The demographic and clinical characteristics of the patients are presented in Table 1. The median age of our study patients was 65.0 (IQR, 58.0–72.0) years and 62.1% of the patients were males. Hepatobiliary cancer (51.7%) was the most frequent cancer type, followed by gastrointestinal cancer (19.0%), gynecologic cancer (10.0%), lung cancer (6.5%), and others (12.8%). The focus of infection significantly differed between the survivor and non-survivor groups. No significant difference in SOFA score was observed between the survivor and non-survivor groups (median, 7.0 vs. 7.0, P = 0.225).

The body compositions in the overall patient population and by sex are presented in Table 2. Muscle depletion was predominant in the overall patient population (87.7%) and did not differ significantly between the survivor and non-survivor groups. Low skeletal muscle attenuation was observed in 42.3% of the patients with statistical difference between the survivor and non-survivor groups (37.8% vs. 48.1%, P = 0.024). The SMIs were significantly higher in survivors compared to that in non-survivors in patients overall (mean, 35.48 vs. 33.32 cm^2^/m^2^, P = 0.002) and in male patients (mean, 37.55 vs. 35.18 cm^2^/m^2^, P = 0.008); however, no significant difference was observed in female patients (mean, 31.91 vs. 30.45 cm^2^/m^2^, P = 0.116). Similarly, the skeletal muscle attenuation was significantly higher in survivors compared to that in non-survivors in patients overall (mean, 35.40 vs. 33.36 HU, P = 0.001) and in male patients (mean, 37.51 vs. 34.72 HU, P < 0.001); however, no significant difference was observed in female patients (mean, 31.73 vs. 31.27 HU, P = 0.613). The clinical characteristics and body composition of the patients with and without muscle depletion are summarized in Appendix A. Patients with muscle depletion were significantly older (median, 66.0 vs. 62.0 years; P = 0.003), more frequently female (male, 58.9% vs. 84.7%; P < 0.001) and had a lower BMI (median, 21.8 vs. 22.8 kg/m^2^; P = 0.029) and skeletal muscle attenuation (mean, 33.80 vs. 39.54 HU; P < 0.001). In multivariable logistic regression analysis, age (adjusted OR, 1.041; 95% CI, 1.012–1.070; P = 0.006), male sex (adjusted OR, 0.277; 95% CI, 0.128–0.600; P = 0.001) and low skeletal muscle attenuation (adjusted OR, 7.454; 95% CI, 2.891–19.219; P < 0.001) were significantly associated with the muscle depletion in cancer patients with septic shock (Appendix A).

In the univariate analysis, the following covariates were significantly associated with 90-day mortality: focus of infection, which was categorized into hepatobiliary, respiratory, and others; SOFA score; SMI; and low skeletal muscle attenuation (Appendix A). Multivariable Cox proportional hazards regression analyses showed that the SMI was independently associated with a lower 90-day mortality (adjusted HR, 0.970; 95% CI, 0.952–0.988; P = 0.001), whereas the presence of low skeletal muscle attenuation was not. For further analysis, the SMI was grouped into quartiles by sex. The multivariable-adjusted HR (95% CI) for 90-day mortality comparing quartiles 2, 3, and 4 of the SMI to the lowest quartile were 0.958 (0.667–1.376), 0.644 (0.438–0.946), and 0.559 (0.373–0.837), respectively (Table 3). A negative association between the SMI quartile and 90-day mortality was consistently observed in male patients, whereas the SMI quartile was not significantly associated with 90-day mortality in female patients. Overall patients and male patients with the SMIs in the first and second quartiles had a significantly shorter survival duration compared to those in overall and male patients with the SMIs in the third and fourth quartiles (Figure 2A,B). However, no significant differences between the SMI quartiles were observed among female patients (P = 0.257 by log-rank test; Figure 2C).

## 4. Discussion

In this registry-based cohort study, we found that muscle depletion was prevalent and that an increased SMI was associated with reduced 90-day mortality in patients with cancer who developed septic shock. The cancer patients had a higher overall prevalence of sarcopenia due to metabolic alterations such as anorexia, hypoanabolism, or hypercatabolism, which might elicit dramatic changes in body composition [26,27]. A recent systemic review reported that the overall prevalence of pre-therapeutic sarcopenia in cancer patients was 40%, ranging from 12% in colorectal cancer to 80% in esophageal cancer [6]. Moreover, patients with sepsis often experience muscle catabolism, muscle weakness, and metabolic dysfunction. Active cancer patients with septic shock can explain the high overall prevalence of muscle depletion (87%), which consisted of pre-sarcopenia and sarcopenia, in our study. Our result is consistent with that of a previous study reporting that the progression of muscle depletion contributed to the development of septic shock in cancer patients. Undernutrition impairs host immune function, especially cell-mediated immunity, including T-lymphocytes, complement activity, phagocytosis, and chemotaxis [28]. When systemic inflammation is accompanied by this state, immune function is further depressed and the patient consequently develops septic shock, a dysregulated response to an infection [28].

The severity of sarcopenia has been shown to be both a marker for overall patient health status and a predictor of patient outcomes [16,29,30]. This is consistent with the findings of the present study. Compared to the lowest quartile of the SMI, higher quartiles of the SMI (third and fourth) were independently associated with a decreased 90-day mortality among the study patients, especially for male patients, after adjusting for other possible confounding factors. 

Body composition and metabolism of fat and protein differ between men and women; however, few studies have assessed the impact of sex differences in body composition on survival in patients with cancer [31,32,33]. Therefore, we analyzed the body composition of our patients by sex. In male patients, the SMA, SMI and skeletal muscle attenuation were significantly higher in survivors than those in non-survivors and the VFA/SFA ratio tended to be lower in survivors than that in non-survivors, although the difference was not statistically significant. In contrast to male patients, body composition, including the SMA, SFA, VFA, and skeletal muscle attenuation, did not differ between survivors and non-survivors in female patients. In overall patients, the SMI was an independent predictor of a lower 90-day mortality after adjusting for other variables that might affect 90-day mortality in septic shock (adjusted HR, 0.970; 95% CI, 0.952–0.988; P = 0.001). However, the association of muscle depletion and 90-day mortality was evident in male but not female patients. 

Sex-based differences have been widely reported for skeletal muscle fiber-type composition, function, and muscle fatigue susceptibility in preclinical models and healthy participants [34,35]. Cancer-induced muscle wasting causes atrophy in type II glycolytic fibers rather than in type I oxidative myofibers [36], and type II glycolytic fibers account for a greater percentage of muscle composition in men than in women [34,37]. Also, men are more susceptible than women to muscle fatigue and this functional impairment was more obvious in male cancer patients than in females [38]. The mechanism of this sexual dimorphism is not yet elucidated despite the acknowledged roles of estrogen and androgen in muscle mass and function [32]. In our real-world study, the prognostic value of body composition, particularly the SMI, differed between male and female patients. Consistent with our findings, Choi et al. also reported that the association between muscle depletion and mortality was more prominent in male compared to female patients with advanced pancreatic cancer [39]. These results imply that muscle depletion, sarcopenia, and cancer cachexia have different progression and prognostic impacts for male and female patients and that prevention and treatment strategies for these pathologic conditions may need to be developed based on the understanding of the properties these sexual dimorphisms [32].

Several limitations should be considered when interpreting the findings of the present study. First, a lack of data regarding more specific cancer-related characteristics such as cancer stage, performance status, and treatment trajectory was a significant limitation. Also, our primary outcome was all-cause 90-day mortality, which did not differentiate causes of mortality, including cancer and septic shock. However, recent studies demonstrated that cancer-related characteristics were not associated with short-term mortality [40,41], and the septic shock registry of our institution excludes patients who refused admission to the intensive care unit and intensive treatment or who signed a “do not attempt resuscitation” order. Considering these exclusion criteria of the septic shock registry, our study patients might have good performance status with a life expectancy of more than three months due to cancer. Second, the treatment of septic shock and cancer differ owing to the long study period accompanied by changes in treatment guidelines. In-hospital treatment strategies such as ventilator use or continuous renal replacement therapy as well as end-of-life decisions could affect the outcome. These potential confounding factors were not standardized between attending physicians. Third, we evaluated the patients using only a measurement of muscle mass adjusted by height squared. The impairment of muscle strength or physical performance such as gait speed and hand-grip strength is another key aspect in patients with sarcopenia and cachexia [11]; however, these data were not available due to the retrospective nature of this cohort study. Finally, this study was performed at the ED of single tertiary referral center in South Korea and all study patients were Asians, which compromises the generalizability of the findings. Race-ethnic differences in body composition have been generally reported [42]; our study adds data on body composition in Asian patients with cancer who developed septic shock. Also, this study included cancer patients with septic shock who underwent an abdominopelvic CT examination at ED presentation. The results of this study were only generalizable to those patients who needed an abdominopelvic CT examination at presentation and cannot be extended to all cancer patients with septic shock.

## 5. Conclusions

In conclusion, in the present study, most patients with cancer who developed septic shock had muscle depletion, which was independently associated with 90-day mortality in patients overall. Only in male patients was a graded association observed between the SMI and 90-day survival. Our results suggest that sex differences should be considered to establish better prognostication and treatment strategies. Further prospective validation studies in this population are warranted.

## Figures and Tables

**Figure 1 jcm-08-01583-f001:**
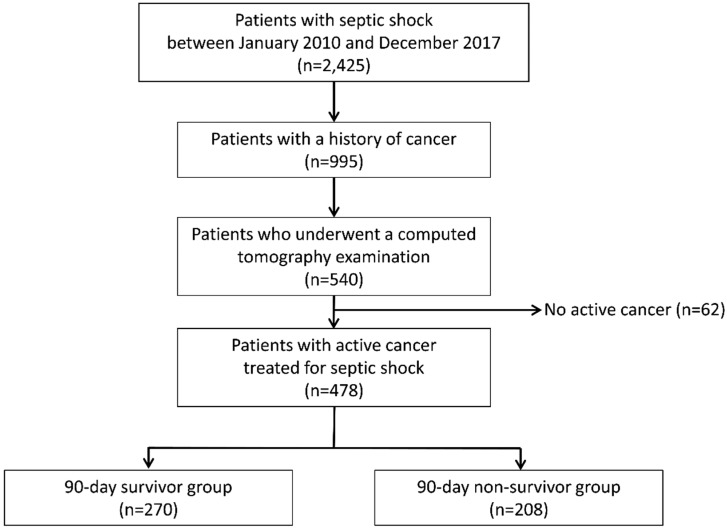
Flow diagram of patient enrolment and allocation in our study.

**Figure 2 jcm-08-01583-f002:**
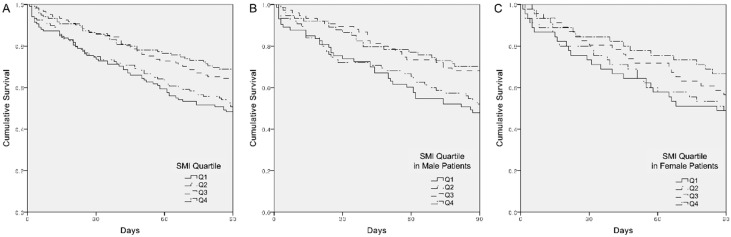
Kaplan–Meier survival curve estimates of 90-day mortality in (**A**) overall, (**B**) male, and (**C**) female patients according to their skeletal muscle index (SMI) quartiles.

**Table 1 jcm-08-01583-t001:** Demographic and clinical characteristics of study patients according to 90-day mortality.

Characteristics	Overall patients (n = 478)	Survivors (n = 270)	Non-survivors (n = 208)	P-Value
Age, years	65.0 (58.0–72.0)	65.0 (58.0–72.0)	65.0 (58.0–72.0)	0.848
Male	297 (62.1%)	171 (63.3%)	126 (60.6%)	0.538
Comorbidities				
Hypertension	169 (35.4%)	94 (34.8%)	75 (36.1%)	0.778
Diabetes mellitus	124 (25.9%)	71 (26.3%)	53 (25.5%)	0.840
Type of solid cancer				0.219
Hepatobiliary	247 (51.7%)	137 (50.7%)	110 (52.9%)	
Gastrointestinal	91 (19.0%)	52 (19.3%)	39 (18.8%)	
Gynecologic	48 (10.0%)	33 (12.2%)	15 (7.2%)	
Lung	31 (6.5%)	13 (4.8%)	18 (8.7%)	
Others	61 (12.8%)	35 (13.0%)	26 (12.5%)	
Focus of infection				0.029
Hepatobiliary	243 (50.8%)	136 (50.4%)	107 (51.4%)	
Respiratory	63 (13.2%)	27 (10.0%)	36 (17.3%)	
Others	172 (36.0%)	107 (39.6%)	65 (31.3%)	
SOFA score	7.0 (5.0–10.0)	7.0 (5.0–9.0)	7.0 (5.0–10.0)	0.225

Data are presented as median (interquartile range), and number (percentage). Abbreviations: SOFA, Sequential Organ Failure Assessment.

**Table 2 jcm-08-01583-t002:** Body composition of study patients by sex and 90-day mortality.

Body Composition	Overall	Survivors	Non-Survivors	P-Value
**Overall patients**				
Number	478	270	208	
BMI, kg/m^2^	22.0 (19.6–24.3)	22.1 (20.0–24.4)	21.7 (19.4–24.1)	0.292
SFA, cm^2^	89.40 (53.71–144.18)	95.14 (59.25–148.25)	85.35 (49.35–140.31)	0.154
VFA, cm^2^	95.29 (56.96–134.44)	100.06 (58.06–140.23)	89.78 (54.16–128.66)	0.212
SMA, cm^2^	89.64 (73.46–107.68)	93.32 (76.67–111.42)	84.88 (71.23–102.26)	0.002
SMI, cm^2^/m^2^	34.54 (7.58)	35.48 (7.54)	33.32 (7.48)	0.002
Skeletal muscle attenuation, HU	34.51 (6.94)	35.40 (6.76)	33.36 (7.00)	0.001
VFA/SFA ratio	1.07 (0.72–1.67)	1.04 (0.69–1.61)	1.13 (0.73–1.73)	0.306
Obesity	90 (18.8%)	50 (18.5%)	40 (19.2%)	0.843
Muscle depletion	419 (87.7%)	231 (85.6%)	188 (90.4%)	0.112
Low skeletal muscle attenuation	202 (42.3%)	102 (37.8%)	100 (48.1%)	0.024
**Male**				
Number	297	171	126	
BMI, kg/m^2^	21.8 (19.4–23.8)	21.9 (19.5–23.7)	21.3 (19.1–23.9)	0.401
SFA, cm^2^	73.65 (44.88–118.68)	78.03 (51.38–118.13)	67.90 (36.83–119.54)	0.133
VFA, cm^2^	100.49 (55.09–148.09)	105.23 (57.61–152.68)	95.75 (53.24–143.99)	0.573
SMA, cm^2^	99.51 (85.99–115.74)	106.01 (90.99–118.78)	93.12 (82.32–110.71)	0.002
SMI, cm^2^/m^2^	36.55 (7.64)	37.55 (7.40)	35.18 (7.79)	0.008
Skeletal muscle attenuation, HU	36.33 (6.75)	37.51 (6.08)	34.72 (7.30)	<0.001
VFA/SFA ratio	1.33 (1.00–1.96)	1.30 (0.93–1.93)	1.42 (1.07–2.03)	0.083
Obesity	45 (15.2%)	24 (14.0%)	21 (16.7%)	0.532
Muscle depletion	247 (83.2%)	138 (80.7%)	109 (86.5%)	0.186
Low skeletal muscle attenuation	105 (35.4%)	51 (29.8%)	54 (42.9%)	0.020
**Female**				
Number	181	99	82	
BMI, kg/m^2^	22.8 (3.9)	23.0 (4.0)	22.6 (3.8)	0.496
SFA, cm^2^	132.92 (75.82)	137.64 (75.26)	127.22 (76.57)	0.359
VFA, cm^2^	92.93 (49.68)	97.95 (54.30)	86.85 (42.99)	0.135
SMA, cm^2^	74.27 (15.61)	75.66 (15.47)	72.59 (15.72)	0.190
SMI, cm^2^/m^2^	31.25 (6.21)	31.91 (6.37)	30.45 (5.96)	0.116
Skeletal muscle attenuation, HU	31.52 (6.17)	31.73 (6.33)	31.27 (5.99)	0.613
VFA/SFA ratio	0.74 (0.52–0.96)	0.75 (0.52–1.00)	0.74 (0.51–0.95)	0.782
Obesity	45 (24.9%)	26 (26.3%)	19 (23.2%)	0.632
Muscle depletion	172 (95.0%)	93 (93.9%)	79 (96.3%)	0.515
Low skeletal muscle attenuation	97 (53.6%)	51 (51.5%)	46 (56.1%)	0.538

Data are presented as mean (standard deviation), median (interquartile range), and number (percentage). Abbreviations: BMI, body mass index; HU, Hounsfield units; SFA, subcutaneous fat area; SMA, skeletal muscle area; SMI, skeletal muscle area index; VFA, visceral fat area.

**Table 3 jcm-08-01583-t003:** Hazard ratios for 90-day mortality by muscle area index quartile in study participants.

Population	HR (95% CI)	P-Value	Multivariable-Adjusted HR (95% CI)	P-Value
**Overall population**				
Q1	Reference		Reference	
Q2	0.920 (0.645–1.312)	0.646	0.958 (0.667–1.376)	0.817
Q3	0.620 (0.423–0.908)	0.014	0.644 (0.438–0.946)	0.025
Q4	0.539 (0.362–0.802)	0.002	0.559 (0.373–0.837)	0.005
**Male**				
Q1 (<31.03)	Reference		Reference	
Q2 (31.03–36.45)	0.894 (0.570–1.402)	0.627	0.950 (0.598–1.511)	0.829
Q3 (36.46–42.06)	0.527 (0.319–0.871)	0.012	0.539 (0.324–0.895)	0.017
Q4 (>42.06)	0.534 (0.323–0.883)	0.014	0.577 (0.344–0.967)	0.037
**Female**				
Q1 (<27.22)	Reference		Reference	
Q2 (27.22–31.35)	0.970 (0.544–1.729)	0.917	0.840 (0.452–1.561)	0.581
Q3 (31.36–35.47)	0.787 (0.435–1.421)	0.426	0.614 (0.314–1.199)	0.153
Q4 (>35.47)	0.549 (0.286–1.053)	0.071	0.418 (0.204–0.856)	0.017

The multivariable model was adjusted for age, sex, hypertension, diabetes mellitus, type of solid cancer (hepatobiliary, gastrointestinal, gynecologic, lung, and others), focus of infection (hepatobiliary, respiratory, and others), and Sequential Organ Failure Assessment score at admission. Abbreviations: HR, hazard ratio; CI, confidence interval; Q, quartile.

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
