# Peer review of "Impact of Body Composition Status on 90-Day Mortality in Cancer Patients with Septic Shock: Sex Differences in the Skeletal Muscle Index"

_jcm, 2019, doi:10.3390/jcm8101583_

Round 1
Reviewer 1 Report
Thank you for the opportunity to review your manuscript; I enjoyed reading about your study and only have a few questions:
Why were GI and respiratory infections combined? Pneumonia is a significant source for sepsis, but from current version of the data we can't see a difference in rates of respiratory infections between groups.
Have you thought to look at how muscle attenuation/average Hounsfield unit is associated with 90-day mortality?
In the limitations, please add that the results of this study are only generalizable to septic patients who needed an abdominal CT scan upon admit. These results cannot be extended to all cancer patients with septic shock.
Author Response
Reviewer #1
Thank you for the opportunity to review your manuscript; I enjoyed reading about your study and only have a few questions:
Why were GI and respiratory infections combined? Pneumonia is a significant source for sepsis, but from current version of the data we can't see a difference in rates of respiratory infections between groups.
Response:We appreciate the reviewer’s comment.The authors categorized the focus of infection of our patients into hepatobiliary, gastrointestinal, respiratory, genitourinary and other organ systems in Table 1. However, when we performed Cox proportional hazards regression analysis, we categorized thefocus of infection into hepatobiliary, respiratory and others, which was combined with gastrointestinal, genitourinary and other infections.
To clarify our data, we revised the focus of infection presented at Table 1 as below:
Table 1.Demographic and clinical characteristics of study patients according to 90-day mortality
|
Characteristics |
Overall patients (n=478) |
Survivors (n=270) |
Non-survivors (n=208) |
P-value |
|
Focus of infection |
0.029 |
|||
|
Hepatobiliary |
243 (50.8%) |
136 (50.4%) |
107 (51.4%) |
|
|
Respiratory |
63 (13.2%) |
27 (10.0%) |
36 (17.3%) |
|
|
Others |
172 (36.0%) |
107 (39.6%) |
65 (31.3%) |
Also, we have added a sentence in results section as below:
“Focus of infection significantly differed between the survivor and non-survivor groups.”
Have you thought to look at how muscle attenuation/average Hounsfield unit is associated with 90-day mortality?
Response: We highly appreciate the reviewer’s recommendation. We totally agree with the reviewer and muscle depletion and low muscle attenuation would be both important prognostic factors in cancer patients with septic shock. We have added the data of muscle attenuation and performed additional analysis per the reviewer’s recommendation. Skeletal muscle attenuation differed between survivors and non-survivors (mean, 35.40 vs. 33.36 HU, P = 0.001). Low skeletal muscle attenuation, defined as <41 HU for BMI <25 kg/m2 and <33 HU for BMI of 25 kg/m2 or more, was significantly associated with a 90-day mortality in univariable Cox proportional hazards regression analyses (HR, 1.41; 95% CI, 1.08-1.86; P = 0.013). However, the association was not observed in multivariable Cox proportional hazards regression analyses. We have added the results in Table 2 and Table S3 and sentences in methods and results section as below.
Table 2.Body composition of study patients by sex and 90-day mortality
|
Body composition |
Overall |
Survivors |
Non-survivors |
P-value |
|
Overall patients |
||||
|
Skeletal muscle attenuation, HU |
34.51 (6.94) |
35.40 (6.76) |
33.36 (7.00) |
0.001 |
|
Low skeletal muscle attenuation |
202 (42.3%) |
102 (37.8%) |
100 (48.1%) |
0.024 |
|
Male |
||||
|
Skeletal muscle attenuation, HU |
36.33 (6.75) |
37.51 (6.08) |
34.72 (7.30) |
<0.001 |
|
Low skeletal muscle attenuation |
105 (35.4%) |
51 (29.8%) |
54 (42.9%) |
0.020 |
|
Female |
||||
|
Skeletal muscle attenuation, HU |
31.52 (6.17) |
31.73 (6.33) |
31.27 (5.99) |
0.613 |
|
Low skeletal muscle attenuation |
97 (53.6%) |
51 (51.5%) |
46 (56.1%) |
0.538 |
Data are presented as mean (standard deviation), median (interquartile range), and number (percentage).
Abbreviations: HU, Hounsfield units
Table S3.Hazard ratios for 90-day mortality by Cox proportional hazards analysis in patients with cancer who developed septic shock
|
Univariable analysis |
Multivariable analysis |
|||
|
HR (95% CI) |
P-value |
HR (95% CI) |
P-value |
|
|
Age, years |
1.00 (0.99–1.01) |
0.991 |
||
|
Male |
0.91 (0.69–1.20) |
0.506 |
||
|
Comorbidities |
||||
|
Hypertension |
1.03 (0.78–1.37) |
0.838 |
||
|
Diabetes mellitus |
1.00 (0.73–1.36) |
0.980 |
||
|
Type of solid cancer |
||||
|
Hepatobiliary |
Reference |
|||
|
Gastrointestinal |
0.97 (0.67–1.39) |
0.854 |
||
|
Gynecologic |
0.66 (0.38–1.13) |
0.127 |
||
|
Lung |
1.51 (0.92–2.49) |
0.105 |
||
|
Others |
1.07 (0.70–1.63) |
0.771 |
||
|
Focus of infection |
||||
|
Hepatobiliary |
Reference |
Reference |
||
|
Respiratory |
1.54 (1.05–2.24) |
0.026 |
1.44 (0.98–2.10) |
0.063 |
|
Others |
0.86 (0.63–1.17) |
0.340 |
0.86 (0.63–1.18) |
0.346 |
|
SOFA score |
1.05 (1.00–1.09) |
0.046 |
1.04 (1.00-1.09) |
0.052 |
|
SMI, cm2/m2 |
0.97 (0.95–0.99) |
0.001 |
0.97 (0.95-0.99) |
0.001 |
|
Low skeletal muscle attenuation |
1.41 (1.08–1.86) |
0.013 |
|
|
Abbreviations: CI, confidence interval; HR, hazard ratio; SMI, skeletal muscle area index; SOFA, Sequential Organ Failure Assessment
Methods2.2. Data collection
The SMA was demarcated using predetermined thresholds (-29 to +190 Hounsfield units [HU]); the VFA and SFA were also demarcated using fat tissue thresholds (-190 to -30 HU) [1]. Skeletal muscle attenuation was assessed as the mean radiodensity in HU of all SMA at L3.
Low skeletal muscle attenuation was defined as <41 HU for BMI <25 kg/m2 and <33 HU for BMI of 25 kg/m2 or more [2].
2.3. Statistical analysis
The model was adjusted for age, sex, and other variables that might affect 90-day mortality: hypertension, diabetes mellitus, type of solid cancer, focus of infection, SOFA score at admission, SMI and skeletal muscle attenuation.
Results
Low skeletal muscle attenuation was observed in 42.3% of the patients with statistical difference between the survivor and non-survivor groups (37.8% vs. 48.1%, P = 0.024). The SMIs were significantly higher in survivors compared to that in non-survivors in patients overall (mean, 35.48 vs. 33.32 cm2/m2, P = 0.002) and in male patients (mean, 37.55 vs. 35.18 cm2/m2, P = 0.008); however, no significant difference was observed in female patients (mean, 31.91 vs. 30.45 cm2/m2, P = 0.116). Similarly, the skeletal muscle attenuation was significantly higher in survivors compared to that in non-survivors in patients overall (mean, 35.40 vs. 33.36 HU, P = 0.001) and in male patients (mean, 37.51 vs. 34.72 HU, P <0.001); however, no significant difference was observed in female patients (mean, 31.73 vs. 31.27 HU, P = 0.613).
In the univariate analysis, the following covariates were significantly associated with 90-day mortality: focus of infection, which was categorized into hepatobiliary, respiratory, and others; SOFA score; SMI; and low skeletal muscle attenuation (Table S3). Multivariable Cox proportional hazards regression analyses showed that SMI was independently associated with a lower 90-day mortality (adjusted HR, 0.970; 95% CI, 0.952-0.988; P = 0.001), whereas presence of low skeletal muscle attenuation was not.
References
[1] Yip C, Dinkel C, Mahajan A, Siddique M, Cook GJ, Goh V. Imaging body composition in cancer patients: visceral obesity, sarcopenia and sarcopenic obesity may impact on clinical outcome. Insights into imaging. 2015;6:489-97.
[2] Martin L, Birdsell L, MacDonald N, et al. Cancer cachexia in the age of obesity: skeletal muscle depletion is a powerful prognostic factor, independent of body mass index. J Clin Oncol. 2013;31:1539-47.
In the limitations, please add that the results of this study are only generalizable to septic patients who needed an abdominal CT scan upon admit. These results cannot be extended to all cancer patients with septic shock.
Response: We appreciate the reviewer’s comment and have added sentences in the limitation section per reviewer’s comments as below:
Also, this study included the cancer patients with septic shock who underwent an abdominopelvic CT examination at ED presentation. The results of this study were only generalizable to those patients who needed an abdominopelvic CT examination at presentation and cannot be extended to all cancer patients with septic shock.

Reviewer 2 Report
Although, the authors argue that the association between sarcopenia and early death in a context of septic shock remains unclear in cancer patients, there is in my opinion sufficient data in the literature between sarcopenia and septic shock, and between sarcopenia and mortality. This is not surprising to show an association between decreasing muscle mass and mortality.
Methods:
This study lacks originality, especially since the selection of cancer patients is an important biais (cancer patients represents only 41% of the cohort as a whole). Again, given the retrospective nature of measurements (i.e. CT-scan), there is a selection bias related to the loss of information about CT-scan measurements.
The authors do not specify in which period the measurement of body composition was made (the last month?). Please specify because body composition is a time-dependent "variable".
The authors do not clearly define sarcopenia in the methods. Indeed, there is no data on what physical performance was used to define sarcopenia (gait speed? hand-grip strength?...). Please specify, and add information in results about physical performances used.
Why the authors did stratified by sex ? Was there an interaction detected ? Or was the assumption of proportional hazards not verified for sex ? Or Was there a multicollinearity ? Please specify because stratified analyses lead to a loss of power.
Results:
It would be interesting to compare patients according to body composition to understand the phenotype associated with low SMI or sarcopenia, more than the comparison between male and female.
Discussion:
Overall, there is a mixture of terms (sarcopenia and SMI) that creates some confusion in discussion. Sarcopenia is the association of impairment in physical performances AND a decreasing of muscle mass. Sarcopenia was not an independent predictor of survival in this study, while SMI was. A decreasing of muscle mass without impairment in physical performances is called "pre-sarcopenia" according to the European Consensus on sarcopenia. This should be clearly state in discussion to lead less confusion.
The authors should provide why sarcopenia was not associated with survival, while decreasing muscle mass was. This is in my opinion the major result of this study.
Author Response
Reviewer #2
Although, the authors argue that the association between sarcopenia and early death in a context of septic shock remains unclear in cancer patients, there is in my opinion sufficient data in the literature between sarcopenia and septic shock, and between sarcopenia and mortality. This is not surprising to show an association between decreasing muscle mass and mortality.
Response: We agree with the reviewer’s comment that many previous studies already demonstrated the association between sarcopenia and mortality in patients with sepsis or in patients with cancer. In this study, we found that muscle depletion was prevalent in cancer patients who developed septic shock (419/478, 87.7%) and the impact of muscle depletion on mortality was different between male and female patients, that the graded association between skeletal muscle index and mortality was only observed in male patients.
Methods:
This study lacks originality, especially since the selection of cancer patients is an important bias (cancer patients represents only 41% of the cohort as a whole). Again, given the retrospective nature of measurements (i.e. CT-scan), there is a selection bias related to the loss of information about CT-scan measurements.
Response: We appreciate the reviewer’s comment and totally agree with the reviewer’s comment on the possibility of selection bias. Among the 995 patients who were enrolled in the septic shock registry and had a history of cancer, 455 (45.7%) patients were excluded due to no abdominopelvic CT examination upon admission. We have added the sentences in limitation section as below:
Also, this study included the cancer patients with septic shock who underwent an abdominopelvic CT examination at ED presentation. The results of this study were only generalizable to those patients who needed an abdominopelvic CT examination at presentation and cannot be extended to all cancer patients with septic shock.
The authors do not specify in which period the measurement of body composition was made (the last month?). Please specify because body composition is a time-dependent "variable".
Response: We appreciate the reviewer’s comment and totally agree with the reviewer’s comments. The body composition including skeletal muscle area, visceral fat area, and the subcutaneous fat area was assessed using the abdominopelvic CT scan which the study patients underwent at their emergency department presentation. We retrieved the data of body weight and height entered into the electronic health records at admission. In our institution, vasopressors such as norepinephrine and epinephrine were administered by weight-based dosing strategy, i.e. mcg/kg/min. Also, our institution recommended setting the tidal volume for the patients receiving mechanical ventilation of 6-8 mL/kg ideal body weight, computed in men as 50 + (0.91 × [height in centimeters − 152.4]) and in women as 45.5 + (0.91 × [height in centimeters − 152.4]). The nurses entered the body weight and height of all the patients with septic shock into the electronic health records at admission and the weight was daily checked for patients in intensive care unit. To clarify the meaning, we have added sentences in methods section as below:
Additional data were collected for this study using the electronic medical records, including weight, height, type of solid cancer, and the presence of abdominopelvic CT scan to evaluate the body composition. We retrieved the data of body weight and height entered into the electronic health records at admission. The body composition, including skeletal muscle area (SMA), visceral fat area (VFA), and the subcutaneous fat area (SFA), was assessed at the L3 vertebral level of abdominopelvic CT scan performed at ED presentation. Abdominopelvic CT images extending from L3 in the inferior direction, which was performed at ED presentation, were assessed.
The authors do not clearly define sarcopenia in the methods. Indeed, there is no data on what physical performance was used to define sarcopenia (gait speed? hand-grip strength?...). Please specify, and add information in results about physical performances used.
Response: We totally agree with the reviewer’s comments. The authors acknowledged the insufficient data on physical performance were one of the major limitations in this study. We defined sarcopenia and low muscle attenuation based on the previous study demonstrated cut-off points associated with survival after optimum stratification in patients with solid malignancies [1]. Based on the definition, sarcopenia was defined as <43 cm2/m2 for BMI <25 kg/m2, <53 cm2/m2 for BMI of 25 kg/m2 or more, and <41 cm2/m2 regardless of BMI for female patients [1]. Although impairment of muscle strength or physical performance is another key aspect in patients with sarcopenia and cachexia, unfortunately the authors could only evaluate the body composition measurement using abdominopelvic CT due to the retrospective nature of the study. We have changed the term “sarcopenia” to “muscle depletion” to clarify the meaning and we also stressed this as a major limitation as below:
2. Materials and Methods
2.1. Study design and population
Sarcopenia was defined as <43 cm2/m2 for BMI <25 kg/m2, <53 cm2/m2 for BMI of 25 kg/m2 or more, and <41 cm2/m2 regardless of BMI for female patients in previous study [1], and due to the lack of the data about muscle strength or physical performance, we defined muscle depletion according to the previous sex-specific, BMI-dependent SMI cutoffs.
4. Discussion
Impairment of muscle strength or physical performance such as gait speed and hand-grip strength is another key aspect in patients with sarcopenia and cachexia [2]; however, these data were not available due to the retrospective nature of this cohort study.
[1] Martin L, Birdsell L, MacDonald N, et al. Cancer cachexia in the age of obesity: skeletal muscle depletion is a powerful prognostic factor, independent of body mass index. J Clin Oncol. 2013;31:1539-47.
[2] Cruz-Jentoft AJ, Baeyens JP, Bauer JM, et al. Sarcopenia: European consensus on definition and diagnosisReport of the European Working Group on Sarcopenia in Older PeopleA. J. Cruz-Gentoft et al. Age ageing. 2010;39:412-23.
Why the authors did stratified by sex ? Was there an interaction detected ? Or was the assumption of proportional hazards not verified for sex ? Or Was there a multicollinearity ? Please specify because stratified analyses lead to a loss of power.
Response: We appreciate the reviewer’s comments. The sex-based differences in body composition is well known, and the definition of sarcopenia is different by sex. The authors performed a sub-group analysis by sex based on the data in the literature, not from statistical point of view.
Results:
It would be interesting to compare patients according to body composition to understand the phenotype associated with low SMI or sarcopenia, more than the comparison between male and female.
Response: We appreciate the reviewer’s recommendations. In this study, the authors aimed to assess the body composition in cancer patients who presented to the emergency department with septic shock state and to evaluate the association between body composition and outcome. We agree with the reviewer’s recommendation that comparison of the clinical characteristics between the patients with and without sarcopenia would be interesting. However, we did not have data about physical performance of the patients, and therefore we have changed the term “sarcopenia” to “muscle depletion” to clarify the meaning. We have performed additional analysis to compare the clinical characteristics between the patients with and without muscle depletion and also evaluated the clinical factors associated with muscle depletion in cancer patients developing septic shock. Age, sex, BMI and skeletal muscle attenuation significantly differed between patients with and without muscle depletion (Table S1). In multivariable logistic regression analysis, age (adjusted OR, 1.041; 95% CI, 1.012-1.070; P = 0.006), male (adjusted OR, 0.277; 95% CI, 0.128-0.600; P = 0.001) and low skeletal muscle attenuation (adjusted OR, 7.454; 95% CI, 2.891-19.219; P <0.001) were significantly associated with the presence of muscle depletion in cancer patients with septic shock (Table S2). We have added these results in our manuscript and supplementary tables, Table S1 and Table S2, as below:
2. Materials and Methods
2.3. Statistical analysis
Descriptive statistics were used to summarize the characteristics of the study patients according to 90-day mortality and according to presence of muscle depletion. Additionally, odds ratios (ORs) and 95% confidence intervals (CIs) for presence of muscle depletion were examined using univariable and multivariable logistic regression analysis. Variables were tested for goodness of fit using variable methods such as Hosmer-Lemeshow test and Stukel test.
3. Results
The clinical characteristics and body composition of the patients with and without muscle depletion are summarized in Table S1. Patients with muscle depletion were significantly older (median, 66.0 vs. 62.0 years; P = 0.003), more frequently female (male, 58.9% vs. 84.7%; P <0.001) and showed lower BMI (median, 21.8 vs. 22.8 kg/m2; P = 0.029) and skeletal muscle attenuation (mean, 33.80 vs. 39.54 HU; P <0.001). In multivariable logistic regression analysis, age (adjusted OR, 1.041; 95% CI, 1.012-1.070; P = 0.006), male (adjusted OR, 0.277; 95% CI, 0.128-0.600; P = 0.001) and low skeletal muscle attenuation (adjusted OR, 7.454; 95% CI, 2.891-19.219; P <0.001) were significantly associated with the muscle depletion in cancer patients with septic shock (Table S2).
Table S1.Demographic and clinical characteristics of study patients with and without muscle depletion
|
Characteristics |
Overall patients (n=478) |
Muscle depletion (n=419) |
No muscle depletion (n=59) |
P-value |
|
Age, years |
65.0 (58.0–72.0) |
66.0 (58.0–73.0) |
62.0 (56.0–67.0) |
0.003 |
|
Male |
297 (62.1%) |
247 (58.9%) |
50 (84.7%) |
<0.001 |
|
Comorbidities |
||||
|
Hypertension |
169 (35.4%) |
147 (35.1%) |
22 (37.3%) |
0.740 |
|
Diabetes mellitus |
124 (25.9%) |
109 (26.0%) |
15 (25.4%) |
0.923 |
|
Type of solid cancer |
0.401 |
|||
|
Hepatobiliary |
247 (51.7%) |
212 (50.6%) |
35 (59.3%) |
|
|
Gastrointestinal |
91 (19.0%) |
79 (18.9%) |
12 (20.3%) |
|
|
Gynecologic |
48 (10.0%) |
46 (11.0%) |
2 (3.4%) |
|
|
Lung |
31 (6.5%) |
28 (6.7%) |
3 (5.1%) |
|
|
Others |
61 (12.8%) |
54 (12.9%) |
7 (11.9%) |
|
|
Focus of infection |
0.280 |
|||
|
Hepatobiliary |
243 (50.8%) |
212 (50.6%) |
31 (52.5%) |
|
|
Respiratory |
63 (13.2%) |
59 (14.1%) |
4 (6.8%) |
|
|
Others |
172 (36.0%) |
148 (35.3%) |
24 (40.7%) |
|
|
SOFA score |
7.0 (5.0–10.0) |
7.0 (5.0–10.0) |
7.0 (5.0–9.0) |
0.836 |
|
90-day mortality |
208 (43.5%) |
188 (44.9%) |
20 (33.9%) |
0.112 |
|
Body composition |
|
|
|
|
|
BMI, kg/m2 |
22.0 (19.6–24.3) |
21.8 (19.4–24.4) |
22.8 (21.0–24.2) |
0.029 |
|
SFA, cm2 |
89.40 (53.71–144.18) |
92.6 4(53.11–144.48) |
87.40 (56.32–139.74) |
0.820 |
|
VFA, cm2 |
95.29 (56.96–134.44) |
94.69 (57.61–138.64) |
97.81 (50.97–124.92) |
0.424 |
|
SMA, cm2 |
89.64 (73.46–107.68) |
84.84 (71.84–98.64) |
125.75 (116.80–134.23) |
<0.001 |
|
SMI, cm2/m2 |
34.54 (7.58) |
32.84 (6.36) |
46.61 (3.44) |
<0.001 |
|
Skeletal muscle attenuation, HU |
34.51 (6.94) |
33.80 (6.80) |
39.54 (5.71) |
<0.001 |
|
VFA/SFA ratio |
1.07 (0.72–1.67) |
1.07 (0.72–1.68) |
1.02 (0.70–1.60) |
0.483 |
|
Obesity |
90 (18.8%) |
81 (19.3%) |
9 (15.3%) |
0.453 |
|
Low skeletal muscle attenuation |
202 (42.3%) |
197 (47.0%) |
5 (8.5%) |
<0.001 |
Data are presented as mean (standard deviation), median (interquartile range), and number (percentage).
Abbreviations: BMI, body mass index; HU, Hounsfield units; SFA, subcutaneous fat area; SMA, skeletal muscle area; SMI, skeletal muscle area index; VFA, visceral fat area.
Table S2.Odds ratios for muscle depletion by logistic regression analysis in patients with cancer who developed septic shock
|
Univariable analysis |
Multivariable analysis |
|||
|
OR (95% CI) |
P-value |
OR (95% CI) |
P-value |
|
|
Age, years |
1.039 (1.013–1.065) |
0.003 |
1.041 (1.012-1.070) |
0.006 |
|
Male |
0.258 (0.124–0.540) |
<0.001 |
0.277 (0.128-0.600) |
0.001 |
|
Comorbidities |
||||
|
Hypertension |
0.909 (0.517–1.598) |
0.740 |
||
|
Diabetes mellitus |
1.031 (0.552–1.928) |
0.923 |
||
|
Type of solid cancer |
||||
|
Hepatobiliary |
Reference |
|||
|
Gastrointestinal |
1.087 (0.537–2.199) |
0.817 |
||
|
Gynecologic |
3.797 (0.882–16.353) |
0.073 |
||
|
Lung |
1.541 (0.434–5.342) |
0.495 |
||
|
Others |
1.274 (0.536–3.024) |
0.584 |
||
|
Focus of infection |
||||
|
Hepatobiliary |
Reference |
|||
|
Respiratory |
2.157 (0.732–6.354) |
0.163 |
||
|
Others |
0.902 (0.509–1.599) |
0.723 |
||
|
SOFA score |
1.017 (0.933–1.109) |
0.706 |
||
|
Low skeletal muscle attenuation |
9.584 (3.758–24.438) |
<0.001 |
7.454 (2.891-19.219) |
<0.001 |
Abbreviations: CI, confidence interval; OR, odds ratio; SMI, skeletal muscle area index; SOFA, Sequential Organ Failure Assessment.
Discussion:
Overall, there is a mixture of terms (sarcopenia and SMI) that creates some confusion in discussion. Sarcopenia is the association of impairment in physical performances AND a decreasing of muscle mass. Sarcopenia was not an independent predictor of survival in this study, while SMI was. A decreasing of muscle mass without impairment in physical performances is called "pre-sarcopenia" according to the European Consensus on sarcopenia. This should be clearly state in discussion to lead less confusion.
Response: We appreciate the reviewer’s recommendations and we totally agree with the reviewer’s comment that there is a mixture of terms between sarcopenia and low muscle mass in our study, which could lead confusion. Sarcopenia is defined as both functional impairment and low muscle mass, however, in this study, we could not differentiate the patient who had low muscle mass and impaired function from those who preserved physical performance despite the low muscle mass due to the lack of the data. To clarify the results, we have revised the word “sarcopenia” to “muscle depletion”. Also, we have revised sentences and expressions which could lead confusion as below:
2. Materials and Methods
2.2. Data collection
Sarcopenia was defined as <43 cm2/m2 for BMI <25 kg/m2, <53 cm2/m2 for BMI of 25 kg/m2 or more, and <41 cm2/m2 regardless of BMI for female patients in previous study [1], and due to the lack of the data about muscle strength or physical performance, we defined muscle depletion according to the previous sex-specific, BMI-dependent SMI cutoffs.
3. Results
Muscle depletion was predominant in the overall patient population (87.7%) and did not differ significantly between the survivor and non-survivor groups. The clinical characteristics and body composition of the patients with and without muscle depletion are summarized in Table S1. Patients with muscle depletion were significantly older (median, 66.0 vs. 62.0 years; P = 0.003), more frequently female (male, 58.9% vs. 84.7%; P <0.001) and showed lower BMI (median, 21.8 vs. 22.8 kg/m2; P = 0.029) and skeletal muscle attenuation (mean, 33.80 vs. 39.54 HU; P <0.001). In multivariable logistic regression analysis, age (adjusted OR, 1.041; 95% CI, 1.012-1.070; P = 0.006), male (adjusted OR, 0.277; 95% CI, 0.128-0.600; P = 0.001) and low skeletal muscle attenuation (adjusted OR, 7.454; 95% CI, 2.891-19.219; P <0.001) were significantly associated with the muscle depletion in cancer patients with septic shock (Table S2).
4. Discussion
In this registry-based cohort study, we found that muscle depletion was prevalent and that increased SMI was associated with reduced 90-day mortality in patients with cancer who developed septic shock.
Moreover, patients with sepsis often experience muscle catabolism, muscle weakness, and metabolic dysfunction. Active cancer patients with septic shock can explain the high overall prevalence of muscle depletion (87%), which consisted of pre-sarcopenia and sarcopenia, in our study.
Impairment of muscle strength or physical performance such as gait speed and hand-grip strength is another key aspect in patients with sarcopenia and cachexia [2]; however, these data were not available due to the retrospective nature of this cohort study.
5. Conclusions
In conclusion, in the present study, most patients with cancer who developed septic shock had muscle depletion, which was independently associated with 90-day mortality in patients overall.
[1] Martin L, Birdsell L, MacDonald N, et al. Cancer cachexia in the age of obesity: skeletal muscle depletion is a powerful prognostic factor, independent of body mass index. J Clin Oncol. 2013;31:1539-47.
[2] Cruz-Jentoft AJ, Baeyens JP, Bauer JM, et al. Sarcopenia: European consensus on definition and diagnosisReport of the European Working Group on Sarcopenia in Older PeopleA. J. Cruz-Gentoft et al. Age ageing. 2010;39:412-23.
The authors should provide why sarcopenia was not associated with survival, while decreasing muscle mass was. This is in my opinion the major result of this study.
Response: We appreciate the reviewer’s comments. As the reviewer commented above, the definition of sarcopenia in our study was only based on Skeletal Muscle Index without consideration of the physical performance. Unfortunately, we could not differentiate the patients with muscle depletion only (pre-sarcopenia) from the patients with muscle depletion and functional impairment (sarcopenia). To clarify the results, we have changed the term “sarcopenia” to “muscle depletion”.

Round 2
Reviewer 2 Report
The article was significantly improved in this version
I do not have any additional comments